# Sentiment Classification of Chinese Tourism Reviews Based on ERNIE-Gram+GCN

**DOI:** 10.3390/ijerph192013520

**Published:** 2022-10-19

**Authors:** Senqi Yang, Xuliang Duan, Zeyan Xiao, Zhiyao Li, Yuhai Liu, Zhihao Jie, Dezhao Tang, Hui Du

**Affiliations:** 1College of Information Engineering, Sichuan Agricultural University, Ya’an 625000, China; 2The Lab of Agricultural Information Engineering, Sichuan Key Laboratory, Ya’an 625000, China; 3Housing and Urban-Rural Development Bureau of Lincheng County, Xingtai 054000, China

**Keywords:** sustainable tourism, tourism satisfaction, scenic spot evaluation text, natural language processing (NLP), text classification

## Abstract

Nowadays, tourists increasingly prefer to check the reviews of attractions before traveling to decide whether to visit them or not. To respond to the change in the way tourists choose attractions, it is important to classify the reviews of attractions with high precision. In addition, more and more tourists like to use emojis to express their satisfaction or dissatisfaction with the attractions. In this paper, we built a dataset for Chinese attraction evaluation incorporating emojis (CAEIE) and proposed an explicitly n-gram masking method to enhance the integration of coarse-grained information into a pre-training (ERNIE-Gram) and Text Graph Convolutional Network (textGCN) (E2G) model to classify the dataset with a high accuracy. The E2G preprocesses the text and feeds it to ERNIE-Gram and TextGCN. ERNIE-Gram was trained using its unique mask mechanism to obtain the final probabilities. TextGCN used the dataset to construct heterogeneous graphs with comment text and words, which were trained to obtain a representation of the document output category probabilities. The two probabilities were calculated to obtain the final results. To demonstrate the validity of the E2G model, this paper was compared with advanced models. After experiments, it was shown that E2G had a good classification effect on the CAEIE dataset, and the accuracy of classification was up to 97.37%. Furthermore, the accuracy of E2G was 1.37% and 1.35% ahead of ERNIE-Gram and TextGCN, respectively. In addition, two sets of comparison experiments were conducted to verify the performance of TextGCN and TextGAT on the CAEIE dataset. The final results showed that ERNIE and ERNIE-Gram combined TextGCN and TextGAT, respectively, and TextGCN performed 1.6% and 2.15% ahead. This paper compared the effects of eight activation functions on the second layer of the TextGCN and the activation-function-rectified linear unit 6 (RELU6) with the best results based on experiments.

## 1. Introduction

According to the 2019 Statistical Bulletin of the National Economic and Social Development of the People’s Republic of China, 6.01 billion tourists visited China in 2019, up 8.4% year-on-year; China’s tourism revenue was 5725.1 billion yuan, up 11.7% year-on-year [1]. Moreover, with the continuous development of tourism, tourists increasingly prefer to check the reviews of attractions before traveling to decide whether to go or not. Visitors’ evaluations of attractions carry strong personal subjective attitudes, and these attraction reviews will be an important reference value for tourists who have not yet visited. In addition, it will have a positive impact on the attractions. A scenic spot can improve its service according to poor comments, further improving tourist satisfaction, and thus improving the economic status of the scenic spot. Attractions will subsequently have more funds to enhance services and improve attraction facilities, creating a virtuous cycle. Highly accurate attraction evaluation classification can also provide technical support for monitoring tourism opinion development and provide basic prerequisites for a series of tasks, such as building knowledge graphs, topic extraction, and other tasks. Therefore, the accurate classification of attraction evaluation is of great importance for tourism development.

In addition, the sentiment classification model E2G for Chinese tourist attraction evaluation provides a good basis for downstream tasks, on which tasks such as building knowledge graphs and theme extraction can be performed to further improve tourist satisfaction.

## 2. Related Work

The deep learning model can classify scenic spot comments. Li et al. used a bidirectional long short-term memory network (BiLSTM) and attention mechanism to learn the sentiment features of travel reviews [2]. They use joint learning to correlate attraction sentiment and attributes with each other for attraction sentiment and attribute prediction. The accuracy of this method on the self-built Shanxi tourism review dataset was improved by 0.98% and 1.54%, respectively, which effectively improved the performance of tourism review sentiment classification. Guo used a tree LSTM model for sentiment analysis of travel website reviews [3]. The experimental results showed that the tree long short-term memory network (LSTM) model had a higher F1 value than Bi LSTM in sentiment classification by 1.19% and had a higher classification ability for tourism reviews [4]. Zhu compared the positive, neutral, and negative effects of the plain Bayesian classification model and the decision tree classification model on the emotional attitude toward tourism in Wutai Mountain [5]. The experimental results showed that good results were achieved on a self-built dataset for the evaluation of tourist attractions in Wutai Mountain. At present, most tourists cannot fully express their emotions with words alone when evaluating the attractions, and more people will choose to go for combining emojis.

A classifier can achieve better results by considering emojis in the dataset during classification. Chen et al. proposed a convolutional self-encoder-based sentiment classification model for microblogs [6]. The model captures the features of emojis by a convolutional self-encoder, then incorporates the features into text features, and then uses a multilayer perception for sentiment classification. He et al. proposed a multichannel convolutional neural network model called emotion-semantics enhanced multi-column deep neural networks (EMCNN) for sentiment semantic enhancement [7], which treats emojis as sentiment words and uses word vectors of emojis to enhance sentiment semantics. Zhao et al. considered not only emojis [8] but also kaomoji in performing sentiment prediction. The embedding of kaomoji, the embedding of emoji, and the word embedding were jointly used as the expression of the text, and then the sentiment was predicted using a context-based bidirectional long short-term memory network (BiLSTM). Jiang et al. mapped emoji and word vectors to a vector space model [9] and used support vector machine (SVM) to implement sentiment classification. It can be seen that incorporating emojis into the dataset can improve the accuracy. However, there is no publicly available travel review dataset that takes emojis into account. In addition, CAEIE addresses this point.

Bidirectional Encoder Representation from Transformers (BERT) is used for emotion classification, which improves the accuracy of emotion classification. Li used BERT to achieve good results for sentiment classification of stock reviews [10]. Yan proposed that a sentiment knowledge-based bidirectional encoder representation from transformers (SK-BERT) considers the emotional tendency of words [11], and the experimental results show that the SK-BERT model outperformed other state-of-the-art models. Wen et al. used BERT to analyze text data containing investors’ emotions [12], which could effectively and accurately identify the emotions contained in the text. However, BERT does not apply to Chinese due to its mask mechanism.

ERNIE does a better job of classifying Chinese datasets. Huang proposed a sentiment classification model called ERNIE-BiLSTM-Attention (EBLA) to classify reviews of goods [13], which achieved more than 0.87 in terms of accuracy, recall, and F1 value. In 2021, Xiao et al. proposed ERNIE-Gram [14], which has a more complete N-gram mask language model. In addition, this model simultaneously establishes the semantic relations between n-grams and achieves learning of the semantic relations between fine and coarse grains simultaneously.

Most of the current text classification methods combine models such as CNN and LSTM [15,16,17], which prioritize sequential and local information and can capture semantic and syntactic information in continuous word sequences very well. However, they ignore global word co-occurrence, which carries discontinuous as well as long-distance semantic information. TextGCN uses the entire dataset to construct a large heterogeneous graph [18]. The model uses a graph convolutional network to jointly learn the embedding of words and the embedding of documents, taking global information well into account. TextGCN is used when classifying text. In 2021, Lin combined BERT and TextGCN to enhance transductive text classification and proposed the BERTGCN model [19]. The experimental results showed a significant effect improvement on five benchmarks of text classification to achieve the current state-of-the-art results. Some scholars have also applied BERTGCN to engineering in various industries. XR et al. used GCN and BERT to perform a joint learning method for text classification and event assignment on the Chinese government hotline [20]. The experimental results showed that the method could achieve better performance. Gao et al. used BERT and GCN [21], named Gating Context Global Network. It was shown experimentally that 0.19%, 0.57%, 1.05%, and 1.17% boosts were obtained on 20NG, R8, R52, and Ohsumed.

The length of tourist comment texts varies widely, with the shortest being only one word and the longest reaching 1000 words. Many scholars are currently working on sentiment classification for attraction review texts. However, more visitors combine emojis when posting comments online. In response to this trend, it is necessary to use emotions as a complementary feature to improve the accuracy of the classification. This will provide a good basis for downstream tasks to further improve tourist satisfaction and promote tourism. In terms of classification models, more scholars utilize models based on BERT and CNN, LSTM, etc., for classification.

However, BERT’s MASK mechanism does not apply to Chinese when feature extraction is performed in Chinese. CNN, LSTM, and other models lack consideration of the overall information of the dataset, although they fully consider the contextual features. These models can only fuse the information before and after the feature in the process of feature extraction and cannot take into account the information of all the data in the whole dataset. That is, a feature can only fuse the information contained in the two pieces of data before and after it, but not the information of the data separated from it directly. In this paper, we combined ERNIE-Gram and TextGCN for modeling [22,23,24]. Feature extraction was performed using ERNIE-Gram for Chinese, taking into full consideration the relationship between coarse-grained and fine-grained data. Coarse and fine granularity refers to the process of performing n-gram masking where the n-gram as a whole is considered as coarse granularity and the n-gram internally is considered as fine granularity. Then, the whole dataset was input to GCN for constructing heterogeneous graphs. ERNIE-Gram fully considers the connections within the data, while GCN fully considers the connections between the data of the whole dataset. Unlike other models that only optimize within individual data, or only optimize globally for the entire dataset, ERNIE-Gram fully considers the relationship between words and makes up for the fact that GCN itself does not consider word order, thus achieving better results. Currently, most of the datasets are for text classification of the evaluations of a single attraction. The CAEIE dataset, however, collects evaluations of different attractions in Beijing, Shanghai, and Tianjin, which greatly increases the richness of the dataset. Meanwhile, the current attraction evaluation dataset does not consider emojis. CAEIE collects and publishes the evaluations incorporating emojis.

The main contributions of this paper include:(1)For the unclear classification of Chinese tourism evaluation, this paper integrated ERNIE-Gram and GCN. ERNIE-Gram was used to make up for the problem of GCN ignoring word order. Thus, ERNIE-Gram considers the features between words and GCN combines the features of global information to further propose the E2G model. Meanwhile, the best results were obtained by performing sentiment classification on a self-built fused expression attraction evaluation dataset.(2)In this paper, we collected the fused expression attraction evaluation dataset by ourselves, and the total dataset reached 44,671. We also cleaned and manually annotated the data after collecting it. We publish CAEIE publicly, filling the problem of there being no public dataset for Chinese attraction evaluation.

## 3. Methods

### 3.1. ERNIE

Sun proposed ERNIE based on the BERT model [25]. BERT when used in Chinese, is based on individual words [26] and ignores word associations, while ERNIE can capture the relationships between words very well. As shown in Figure 1, when BERT is used, it randomly masks 15% of the text. BERT does not consider contextual connections, resulting in a word being separated and not easily calculated for the masked text. The masking strategy of ERNIE considers the relationship between words and masks them with words, making it easier to reason about the masked words. A comparison of the masking methods for BERT and ERNIE is shown in Figure 1.

Taking ‘This place is suitable for cycling around in spring.’ as an example, BERT and ERNIE have different masking strategies. ERNIE incorporates external knowledge in the mask, which makes it easier to calculate the masked part [25]. The core part of ERNIE is the transformer-encode, as shown in Figure 2.

Figure 2 shows the encoding part of the Transformer. The input data is encoded. The data is computed using a multi-headed attention mechanism after adding location information to the data. Think of normalization, forward propagation, and normalization as one layer, and after N such layers, the encoded data is output.

### 3.2. ERNIE-Gram

BERT’s language modeling focuses on the representation of fine-grained text units, with little consideration of coarse-grained linguistic information, resulting in insufficient features. However, ERNIE-Gram is capable of learning both fine-grained and coarse-grained semantic information. The mask mechanism is shown in Figure 3.

As shown in (a) [14], the masked consecutive n-gram words X_2_ and X_3_ correspond to the predicted masked word X in the following equation. The Z_2_ after passing the mask corresponds to $Z_{\backslash M}$ in the following Equation. The loss function is as follows, and $Z_{M}$ denotes the masked word:(1)$$-\log p\theta (Z_{M}|Z_{\backslash M})=-\underset{Z\in Z_{M}}{\sum }\underset{X\in Z}{\sum }\log p\theta (X|Z_{\backslash M})$$

As shown in (b), ERNIE-Gram uses an explicit n-gram representation with Y_2_ as the target for training, reducing the prediction space of the n-gram. In Equation (2), ${\bar{Z}}_{\backslash M}$ denotes the masked sentence and $Y_{M}$ denotes the masked word.
(2)$$-\log p\theta (Y_{M}|{\bar{Z}}_{\backslash M})=-\underset{Y\in Y_{M}}{\sum }\log p\theta (Y|{\bar{Z}}_{\backslash M})$$

As shown in (c), ERNIE-Gram predicts n-grams at both fine and coarse granularity, which helps to extract comprehensive n-gram semantics. The loss function is shown in Equation (3).
(3)$$-\log p\theta (Z_{M}|Z_{\backslash M})=\;-\underset{Y\in Y_{M}}{\sum }\log p\theta (Y|{\bar{Z}}_{\backslash M})-\underset{Z\in Z_{M}}{\sum }\underset{X\in Z}{\sum }\log p\theta (X|{\bar{Z}}_{\backslash M})$$

### 3.3. TEXTGCN

TextGCN uses graph convolutional network models for text models [18]. The model constructs a heterogeneous graph of the dataset based on word co-occurrence and the relationship between text words, and then TEXTGCN learns the representation of text for text classification. Traditional word embeddings use word embeddings such as word2vec, glove and CNN, and RNN, but TEXTGCN can be used for the latest text classification methods without additional word embeddings or prior knowledge.

TEXTGCN learns both word and document embeddings by heterogeneous graphs for text classification. First, TEXTGCN constructs a heterogeneous graph using the documents and words in the dataset, where the documents and words are nodes. After constructing the graph, TextGCN uses the graph convolutional network to learn more comprehensive features for the representation of the nodes. In the classification task, these updated representations can be fed into the classifier for text classification.

As shown in Figure 4, the number of nodes in the figure is the sum of the words and documents in the dataset. The nodes starting with O are document nodes and the other nodes are word nodes. The black lines in the figure represent the edges between documents–words, and the gray ones indicate the edges between words–words. R(x) denotes the embedding representation of x (including the embedding of documents and words). The different colors of the nodes represent different types of documents. Where there are word-to-word edges and word-to-document edges, both types of edges are constructed in the following manner.

(1)The construction method of the edge between words and documents is as follows: Decide whether to create a word–document edge based on whether the word appears in a certain document for a certain word or not. If a word appears in a document, it is linked, and if a word does not appear in a document, it is not linked. The weights of the edges between documents and words are represented by the term frequency–inverse document frequency (TF-IDF), and the word frequency TF indicates how often the keyword appears in the text. As shown in Equation (4).(4)$$tf_{ij}=\frac{n_{i,j}}{\sum_{k}n_{k,j}}$$
where $n_{i,j}$ is the number of occurrences of the word in file $d_{j}$ and the denominator is the sum of the occurrences of all words in file $d_{j}$.

The IDF indicates the IDF of a particular word and can be obtained by dividing the total number of documents by the number of documents containing the word and taking the quotient as a logarithm. The fewer the documents containing the word, the larger the IDF, and the better the category differentiation ability of the word.
(5)$$idf_{i}=log\frac{\left|D\right|}{\left|\left\{j:t_{j}\in d_{j}\right\}\right|}$$
where $\left|D\right|$ is the total number of files in the dataset. $\left|\left\{j:t_{j}\in d_{j}\right\}\right|$ indicates the number of files containing the word $t_{j}$. If the word is not in the corpus, it leads to a denominator of zero. Therefore, in general, 1+$\left|\left\{j:t_{j}\in d_{j}\right\}\right|$ is used. The TF-IDF is represented as shown in Equation (6).
(6)TF−IDF=TF∗IDF


(2)The construction method of word-to-word connection is as follows:


The word–word edges are based on global word co-occurrence information. The global word co-occurrence information is slid across the entire dataset using a fixed-size sliding window to count the word co-occurrence information, and then the weights of the two-word node links are calculated using point mutual information (PMI). The details are as follows.
(7)$$\mathrm{PMI}\left(i,j\right)=\log\frac{p\left(i,j\right)}{p\left(i\right)p\left(j\right)}$$
(8)$$p\left(i,j\right)=\frac{\#W\left(i,j\right)}{\#W}$$
(9)$$p\left(i\right)=\frac{\#W\left(i\right)}{\#W}$$
where #*W* denotes the total number of sliding windows, $\#W\left(i\right)$ denotes the number of sliding windows that contain word *i*, and $\#W\left(i,j\right)$ denotes the number of sliding windows that contain both word *i* and word *j*. After the statistics, the PMI is calculated by bringing in the above equation. A positive PMI means that the semantic relevance between words is high, and a negative one means that the semantic connection between two words is small or non-existent, so we only connect the edges to the two-word nodes with positive PMI, and the weights on the edges are the PMI values of the two words. The adjacency matrix of the heterogeneous map is defined as shown in Equation (10).
(10)$$A_{ij}=\left\{\;\begin{array}{c} \mathrm{PMI}\left(i,j\right)\;i,j\;\mathrm{are}\;\mathrm{words},\;\mathrm{PMI}\left(i,j\right)>0\;\\ \mathrm{TF}-{\mathrm{IDF}}_{ij}\;i\;\mathrm{is}\;\mathrm{document},j\;\mathrm{is}\;\mathrm{word}\;\\ 1\;i=j\;\\ 0\;\mathrm{otherwise}\; \end{array}\right.$$

A two-layer graph convolutional network is used, with the following structure.
(11)$$Z=\mathrm{softmax}\left(\tilde{A}\;\mathrm{ReLU}\left(\tilde{A}XW_{0}\right)W_{1}\right)$$

The first layer uses ReLU as the activation function and the second layer uses the SoftMax function for classification. The loss function is shown in (12).
(12)$$L=-\underset{d\in yD}{\sum }\overset{F}{\underset{f=1}{\sum }}Y_{df}\ln Z_{df}$$

### 3.4. BERTGCN

The current mainstream approach to solving transductive text classification is to use graph networks [27,28] such as GNN, GCN, and GAT. All the labeled and unlabeled data are constructed in a graph, and the nodes in the graph represent documents or words. Through information transfer between the nodes, the model can infer the features of unlabeled nodes in the heterogeneous graph with the information of the labeled nodes, and thus achieve transductive classification. Multilayer GCNs for short text classification can lead to over-smoothing of node features, causing local features to converge to an approximation, resulting in degraded classification performance. However, BERTGCN combines BERT with GCN for transductive text classification to avoid such problems. In addition, the memory storage mechanism dynamically updates a small number of document nodes with each iteration. It avoids reading all the features into BERT for computation at once, which greatly reduces the memory overhead.

The joint BERT module and GCN module is built on top of the BERT module. Training and using BERTGCN consists of one or three main steps.
The construction of heterogeneous graphs and the initialization of document nodes with the BERT model.The joint training of the BERT module with the GCN module.Inference using the trained BERTGCN.

The first step of BERTGCN is to construct a heterogeneous graph consisting of nodes and edges.

There are two types of nodes: word nodes and document nodes, and edges connect between words and between words and documents. The weights of the edges are determined by TF-IDF and PPMI as shown in Equations (6) and (7). BERTGCN initializes all document nodes $X_{\mathrm{doc}}$ with BERT and initializes all word nodes *X* to 0, as shown in (14).
(13)$$X={\left(\begin{array}{c} X_{\mathrm{doc}}\\ 0 \end{array}\right)}_{\left(n_{\mathrm{doc}}+n_{\mathrm{word}}\right)\times d}$$

A GCN model is constructed using the feature vectors of the nodes, and after multiple graph propagation, the output of the features from the last layer of the GCN is used as the input of SoftMax to obtain the distribution of the categories, as shown in Equation (15).
(14)$$Z_{GCN}=\mathrm{softmax}\left(g\left(X,A\right)\right)$$
where $g\left(X,A\right)$ is the graph model. The model is trained using the standard cross-entropy loss function.
(15)$$Z_{\mathrm{BERT}}=\mathrm{softmax}\left(WX\right)$$

BERTGCN activates with SoftMax by feeding the document embedding directly into a dense layer, where *X* denotes the document.

BERTGCN performs linear interpolation for GCN prediction and BERT prediction, as shown in Formula (17).
(16)$$Z=\lambda Z_{\mathrm{GCN}}+\left(1+\lambda \right)Z_{\mathrm{BERT}}$$

When λ = 1, the BERT module is not updated; when λ = 0, the TEXTGCN module is not updated; and when λ ∈ (0,1), both modules get updated and fast convergence of the overall BERTGCN module is achieved by adjusting λ.

## 4. E2G Model (Based on ERNIE-Gram and GCN)

### 4.1. E2G Framework Diagram

To improve the accuracy of Chinese travel review classification, this paper proposed that the E2G model refers to the BERTGCN model, and the model is trained by the joint training of ERNIE-Gram and TextGCN. ERNIE-Gram’s mask mechanism fully considers the relationship between coarse-grained and fine-grained data, which alleviates the problems of varying length and inadequate features in Chinese travel review texts. However, ERNIE-Gram fails to make a global consideration, while GCN makes up for this shortcoming well, thus combining ERNIE-Gram and GCN can achieve good results. The overall framework is shown in Figure 5. The whole model consists of four main stages, including a pre-processing stage, ERNIE-Gram stage, GCN stage, and calculation stage.
(1)Pre-processing stage

The Chinese travel evaluation text is inputted and then pre-processed with data. The text is subjected to the removal of meaningless words. However, we retained the emojis. After removing the meaningless words, they are then input to ERNIE-Gram and TextGCN.


(2)ERNIE-Gram stage


The pre-processed text is fed into ERNIE-Gram, which uses the transformer coding layer. In addition, its better consideration of internal connections leads to output features with richer semantics.


(3)GCN stage


The text is input into TextGCN for constructing heterogeneous graphs and establishing word nodes and document nodes. The edges between the nodes are established using Equation (10), and the trained TextGCN is obtained by continuous iteration. TextGCN considers the whole dataset, which is a good remedy for the lack of global consideration when using ERNIE-Gram. The experimental results showed that E2G uses ReLU6 as the activation function of the second layer of the GCN, and the formula is as in (18), where X is the node, and the specific formula is shown in (14).
(17)$$Z=\mathrm{softmax}\left(\tilde{A}\;\mathrm{ReLU}6\left(\tilde{A}XW_{0}\right)W_{1}\right)$$


(4)Calculation stage


After obtaining the respective features of ERNIE-Gram and TextGCN, the final probabilities are calculated using Equation (17).

### 4.2. E2G Flow Chart

The overall process of E2G is shown in Figure 6, where the text is input into ERNIE-Gram and GCN, respectively, after pre-processing. ERNIE-Gram can achieve rich features by using transformer-encode and then sending to SoftMax. GCN constructs a heterogeneous graph from the input text dataset. The nodes with the white background are words and the nodes with the blue and orange background document. After passing them through the hidden layer, the representations of words and documents are obtained, and thus SoftMax is performed. After obtaining the probabilities of the two parts, the probabilities obtained from ERNIE-Gram are multiplied with m (m is assumed to be 0.2 in the figure), and the probabilities obtained from GCN are multiplied by (1-m) to obtain the final probabilities.

## 5. Experimental Part

### 5.1. Dataset

There is no publicly available dataset about Chinese fusion emoji reviews of tourist attractions. In this paper, we collected more than 40,000 datasets from the Ctrip website, mainly collected for the reviews of Badaling Great Wall, Forbidden City, Disneyland, Beijing Zoo, Tianjin Eye, and other attractions in the past year. The dataset was processed to consist of four categories, including positive and negative reviews without emojis and positive and negative reviews with emojis. The pre-processing process is shown in Figure 7.

After processing, we found that although there were many emojis, some of them could not express specific feelings, such as some plants, animals, and other emojis. We added emojis to some good and bad reviews in reasonable positions. The main reason was that the collected dataset contained some emojis that were unique to Android or Apple systems and could not be displayed in Windows and Linux for encoding and decoding. We added an emoji that can be displayed to the original non-displayable symbols (expressing the same sentiment). In addition, another part was that we observed the location of emojis in real-life evaluations and found that most of them were at the end of a sentence and that people will type more than one symbol when they are happy or angry, so we expanded the dataset appropriately. In the collected dataset, medium reviews were discarded, but when including medium reviews, plain text and reviews containing emojis could reach roughly 9:1 in the overall dataset. After discarding medium reviews, a large number of reviews with emojis were discarded, and we added them manually in order to compensate for this part and maintain the overall ratio at 9:1. The main added emojis are shown in Table 1.

We added the symbolic expressions in Table 1 to part of the dataset. Since people will add more than one expression when they are extremely happy and angry, we added one to five expressions randomly, and the dataset after adding the expressions is shown in Table 2.

Table 2 shows the presentation of the dataset, which was divided into four subcategories for presentation, namely, reviews containing emojis and reviews without emojis. Each category contained positive and negative reviews, and for the convenience of understanding, this paper translated the Chinese in the table into English for display purposes. To be more intuitive, this paper made word clouds of positive and negative reviews according to their word frequencies. The effect is shown in Figure 8.

Observing Figure 8, it is easy to conclude that the keywords for good reviews in (a) are: not bad, go again, recommendable, worth, etc. In (b), the keywords for bad reviews are line up, service, guide, hour, poor experience, etc.

To show the overall data distribution of CAEIE, the overall number of data and the number of individual categories are shown in Table 3.

As shown in Table 3, the dataset was counted in this paper, and there were 36,197 bad reviews without emojis and 5611 positive reviews, 262 bad reviews with emojis, and 2601 positive reviews. The training set, test set, and validation set were divided according to the total number of 8:1:1, and there were 35,735, 4468, and 4468 items, respectively, in total.

In order to show the distribution of CAEIE lengths, the distribution of lengths is calculated and plotted in Figure 9.

To demonstrate the diversity of CAEIE, information entropy was calculated for the four categories () and the overall dataset in this paper, and the results are shown in Figure 10.
(18)$$Hx=-\underset{x\in X}{\sum }p\left(x\right)\log p\left(x\right)$$

x denotes the random variable. $p\left(x\right)$ denotes the output probability function. The greater the uncertainty of a variable, the greater the entropy.

### 5.2. Experimental Details

#### 5.2.1. Experimental Environment

This paper used the PyTorch development framework for experiments, with an Intel Xeon Gold 5218 CPU, a v100 GPU, and 32G of video memory.

#### 5.2.2. Parameter Setting

The control variables method was used to find the best effect of m, GCN-layer, and Dropout.

To test the relationship between m values and accuracy, nine sets of experiments were conducted in this paper to compare the different accuracy rates corresponding to different m, as shown in Figure 11.

In order to test the relationship between Layer and accuracy, six sets of experiments were conducted in this paper, and 1–6 were selected for comparison, and the corresponding accuracy values were obtained, as shown in Figure 12.

To test the relationship between Dropout and accuracy, five sets of experiments were conducted in this paper, and five typical sets of values were selected to obtain the corresponding accuracy values, as shown in Figure 13.

After the experiments, this experiment set m = 0.2, GCN-layer = 2, and dropout = 0.1, as well as a maximum sentence length of 128, batch size of 256, an epoch of 50, hidden layer of 200, and a learning rate of 10^−3^ for the GCN part and 10^−5^ for ERNIE-Gram.

### 5.3. Evaluation Strategy

The model evaluation criterion used in this paper was the accuracy rate (ACC) [30,31,32]. The model evaluation metrics of classification algorithms are often measured by confusion matrices, as shown in Table 4.

In Table 4, TP indicates that it is a positive sample, and the classification result is positive; TN indicates that it is a negative sample, and the classification result is negative; FP indicates that it is a positive sample, and the classification result is negative; FN indicates that it is a negative sample, and the classification result is positive. The accuracy was calculated using four data, and it was the core metric for evaluating the classification models. The accuracy of the classification model (accuracy) indicates the proportion of samples correctly predicted by the model to all samples, and in general, the higher the accuracy, the better the classification, as calculated in (19).
(19)$$\mathrm{Accuracy}=\frac{\mathrm{TP}+\mathrm{TN}}{\mathrm{TP}+\mathrm{FP}+\mathrm{TN}+\mathrm{FN}}$$

The precision of the classification model is defined as the percentage of samples with true positive class among all samples predicted to be positive class, and the formula is as follows:(20)$$\mathrm{Precision}=\frac{\mathrm{TP}}{\mathrm{TP}+\mathrm{FP}}$$

The recall of a classification model is defined as the percentage of samples with true positive classes that are correctly predicted, and the formula is as follows:(21)$$\mathrm{Recall}=\frac{\mathrm{TP}}{\mathrm{TP}+\mathrm{FN}}$$

F1-score is the summed mean of precision and recall. A higher value of F1 indicates a better model prediction.
(22)$$F1=\frac{2\mathrm{TP}}{2\mathrm{TP}+\mathrm{FP}+\mathrm{FN}}$$

### 5.4. Experiments

In this paper, several models were chosen for comparison, including BERT, as well as the more advanced robustly optimized BERT (Roberta) [33], and the BERT-based improvements in ERNIE, as well as TEXTGCN and TEXTGAT. BERT is used for pre-training to learn the embedding representation of a given word in a given context [34]. Roberta builds on BERT by modifying the key hyperparameters in BERT, removing the next sentence pre-training target of BERT and training with a larger batch size and learning rate. Roberta takes an order of magnitude longer to train than BERT. A lite BERT for self-supervised learning of language representations (ALBERT) is also based on BERT’s model [35]. The main differences are the change in the sentence prediction task to predicting coherence between sentences, the factorization of embedding, and the sharing of parameters across layers. Sentiment Knowledge Enhanced Pre-training for Sentiment Analysis (Skep) is based on Roberta’s model. Unlike Roberta, Skep uses a dynamic mask mechanism. The model is more focused on emotional knowledge to build pre-trained targets, which in turn allows the machine to learn to understand emotional semantics.

ERNIE obtained state-of-the-art results on natural language processing (NLP) tasks in Chinese based on the BERT model. ERNIE improved the mechanism of the mask. Instead of the basic word piece mask, its mask added external knowledge in the pretraining stage. ERNIE-Gram is based on ERNIE and refined the mask mechanism. The model established semantic relationships between n-grams synchronously and achieves simultaneous learning of fine and coarse grains.

TextGCN is based on GCN to build a heterogeneous graph of the dataset, treating words and documents as nodes, which were continuously trained to obtain a representation of the nodes. The current optimal results were obtained using Roberta combined with GCN on five typical datasets—20 Newsgroups, R8, R52, Ohsumed, and Movie Review.

As shown in Table 5, the E2G model achieved the best results on the CAEIE dataset compared to several other comparable models. E2G had an accuracy lead of 1.37% and 1.35% compared to ERNIE-Gram and TextGCN, respectively. The reason was that although ERNIE-Gram achieved good results and ERNIE-Gram fully considered the connection between n-grams when extracting features, it lacks global considerations. On the contrary, although TextGCN fully considers the relationship of the whole dataset, it ignores the word order and lacks consideration between n-grams. Incorporating ERNIE-Gram is just able to solve this problem. In addition, in order to verify the stability of E2G, this paper used the ten-fold crossover method for validation, and the error of the obtained accuracy results did not exceed 0.001.

To demonstrate the effectiveness of E2G, we performed ablation experiments that tested the combination of GAT in ERNIE and ERNIE-Gram. GAT is a neural network based on graph-structured data. The model addresses the shortcomings of GCN using a hidden self-attentive layer.

As shown in Table 6, ERNIE and ERNIE-Gram combined with GAT lagged 1.44% and 0.62% behind the effect of combining with GCN, respectively. Although both GCN and GAT aggregate features from neighboring vertices to the center vertex, GCN utilizes the Laplace matrix while GAT utilizes attention coefficients. To some extent, GAT is stronger, but the experimental results proved that, in textual data, the use of fixed coefficients was more applicable, which was also confirmed in BERTGCN.

To verify the effects of different activation functions on GCN, this paper tried to use different activation functions in the second layer of GCN. The main ones used for the experiments included rectified linear unit (RELU), ReLU6, exponential linear unit (ELU), scaled exponential linear unit (Selu), continuously differentiable exponential linear units (CELU), leaky rectified linear units (Leaky ReLU), randomized rectified linear units (RReLU), and gaussian error linear units (Gelu).

The ReLU function is a segmented linear function [36] that changes all negative values to 0, while positive values remain unchanged. In the case that the input is negative, it will output 0, and then the neuron will not be activated. This means that only some of the neurons will be activated at the same time, thus making the network very sparse, which is very efficient for computation.

ReLU6 is the ordinary ReLU but restricts the maximum output value to 6 [37], and Relu uses x for linear activation in the region where x > 0, which may cause the value after activation to be too large and thus affect the stability of the model.

ELU solves the problem that exists with ReLU [38]. ELU has a negative value, which causes the average value of activation to approach zero and thus speeds up learning. At x < 0, the gradient is 0. This neuron and subsequent neurons have a gradient of 0 forever and no longer respond to any data, resulting in the corresponding parameters never being updated.

Selu after this activation function makes the sample distribution automatically normalized to 0 mean and unit variance [39]. Selu’s positive semi-axis is greater than 1, which allows it to increase when the variance is too small and prevents the gradient from vanishing. The activation function then has an immobility point. The output of each layer has a mean of 0 and a variance of 1.

CELU is similar to SELU [40], in that CELU uses negative intervals for exponential computation and integer intervals for linear computation. CELU facilitates the convergence and generalization performance of neural networks.

Leaky ReLU is a variant of the ReLU activation function [41]. The output of this function has a small slope for negative-valued inputs. Since the derivative is always non-zero, this reduces the appearance of silent neurons. Moreover, allowing gradient-based learning solves the ReLU function after it enters the negative interval.

RReLU is a variant of Leaky ReLU [42]. In RReLU, the slope of the negative values is random in training and becomes fixed in testing.

GELU achieves regularization by multiplying the input by 0 or 1 [43], but whether the input is multiplied by 0 or 1 depends on the random choice of the input’s distribution.

As shown in Table 7, ReLU6 achieved the best results on the CAEIE dataset when m = 0.2, layer = 2, and dropout = 0.1. ReLU6 led by 0.126 and 0.027 in F1 and accuracy, respectively. ReLU6 was essentially the same as ReLU but limited the maximum output value to 6. If there is no restriction on the activation range of ReLU, the output ranges from 0 to positive infinity. If the activation values are very large and distributed over a large range, it is not possible to describe such a large range of values well and accurately, resulting in a loss of accuracy. ReLU6 avoids this problem well and obtained the highest accuracy.

## 6. Conclusions

In this paper, we classified the sentiment of visitors’ evaluations of attractions, integrated emojis into the comments, and made the collected dataset public to fill the gap in the Chinese integration of emoji attraction evaluation. In addition, this paper proposed the E2G model, which firstly preprocesses the text and then sends it to ERNIE-Gram and TextGCN, respectively. ERNIE-Gram is trained using its unique mask mechanism to obtain the final probability. TextGCN uses the dataset to construct a heterogeneous graph with documents and words as nodes, and finally obtains a representation of the nodes and outputs the category probabilities. The two probabilities are calculated to obtain the final result. The experiment showed that E2G had a good classification effect on Chinese attraction evaluation. The accuracy of classification was up to 97.37% and, compared with ERNIE-Gram, the TextGCN and E2G accuracy was ahead by 1.37% and 1.35%, respectively. In this paper, the evaluation was divided into three categories, 1–2 as bad reviews, 4–5 as good reviews, and 3 as medium reviews, and only good and bad reviews were used in the process of constructing the dataset, and no consideration of medium reviews was carried out. The reason was that we observed that the score was given 3 points despite the bad content of many visitors’ comments. This made our classification work very difficult and greatly affected the effectiveness of the classification. In the future, special treatment can be given to moderate reviews, thus allowing for the three categories of bad, moderate, and good reviews. The E2G model required a lot of video memory during training. In the future, the algorithm can be optimized to reduce the time and memory required for training, thus improving the practicality. In addition, there were some visitors in the comments who liked to use irony to express their dissatisfaction. This phenomenon reduced the accuracy of text classification and ironic comments can be specifically identified in the future.

## Figures and Tables

**Figure 1 ijerph-19-13520-f001:**
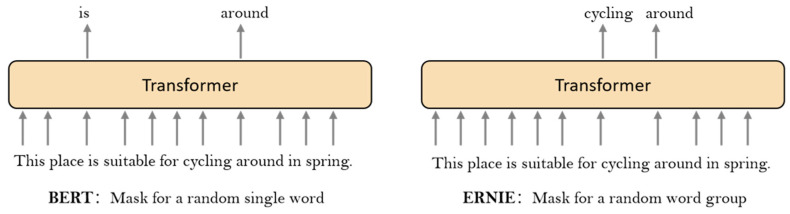
Comparison of BERT and ERNIE mask mechanisms.

**Figure 2 ijerph-19-13520-f002:**
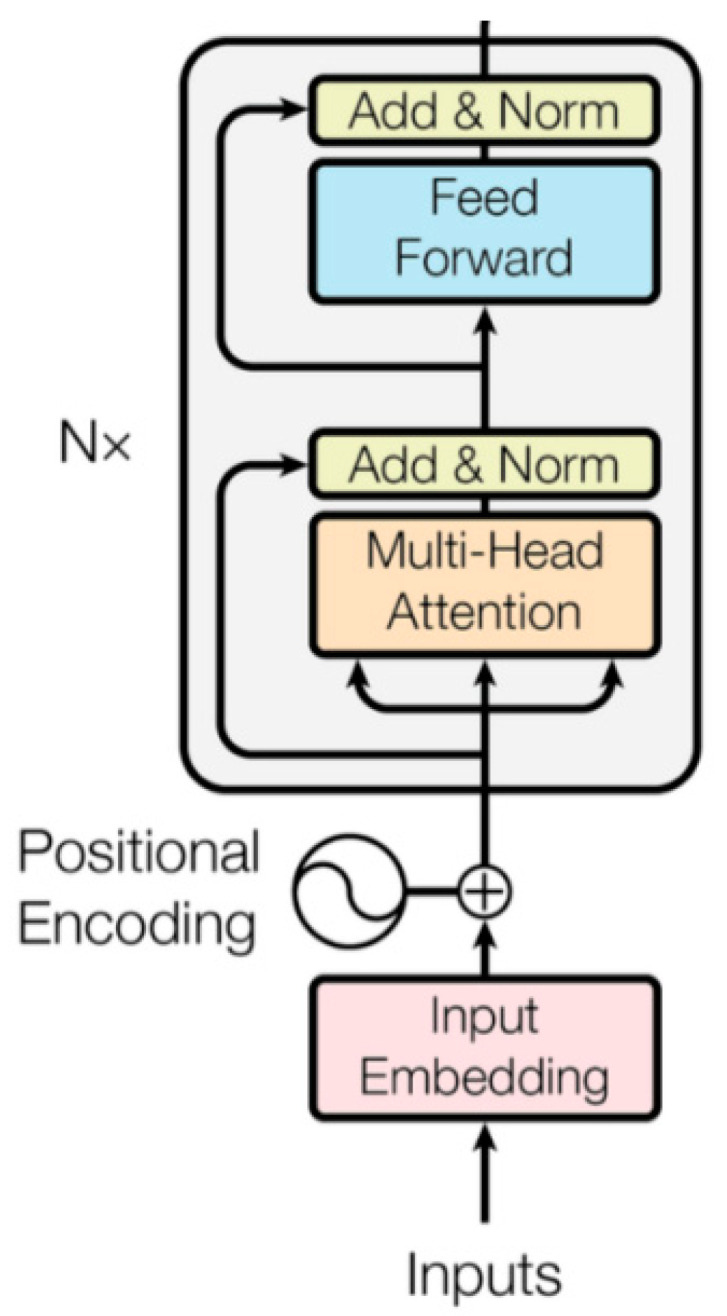
Framework diagram of transformer-encode.

**Figure 3 ijerph-19-13520-f003:**
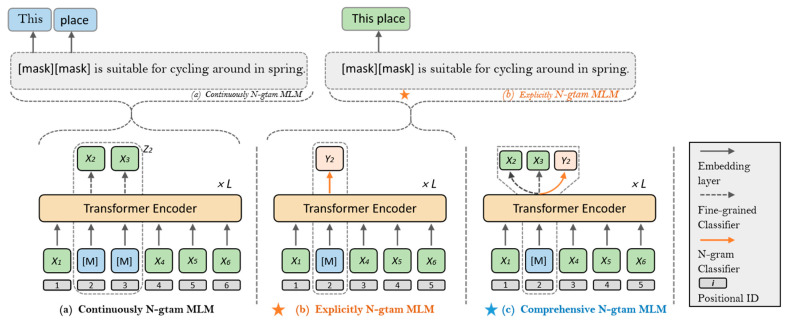
Comparison of ERNIE-Gram and ERNIE mask mechanisms.

**Figure 4 ijerph-19-13520-f004:**
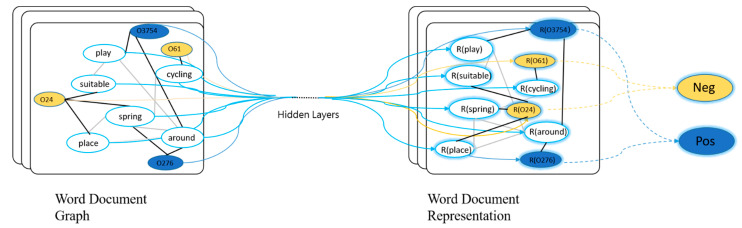
Schematic of TextGCN.

**Figure 5 ijerph-19-13520-f005:**
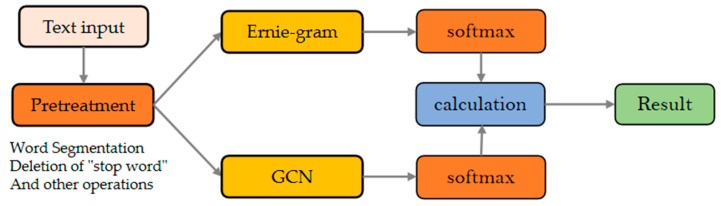
Structure diagram of EGC.

**Figure 6 ijerph-19-13520-f006:**
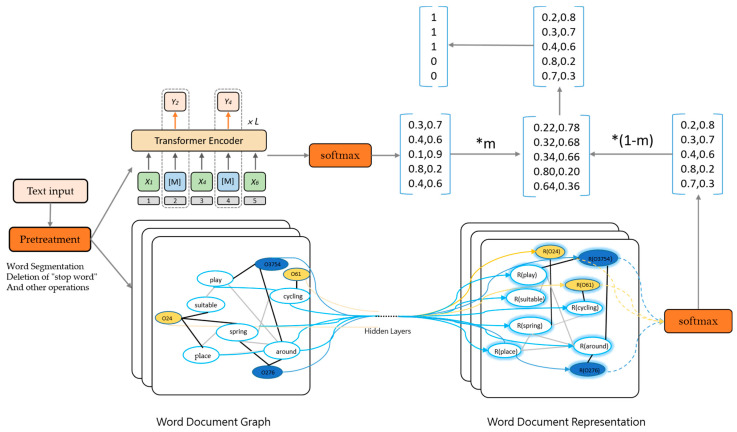
E2G flow chart. * implies multiplication.

**Figure 7 ijerph-19-13520-f007:**
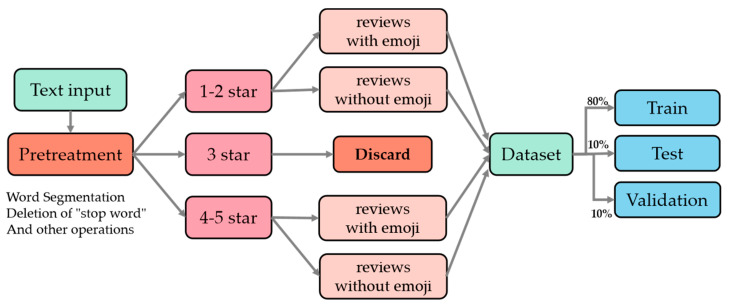
Data preprocessing flow chart. After the text was input, it was preprocessed to classify 1–2 stars as poor ratings and 4–5 stars as good ratings. A rating of 3 stars was a medium rating, which was not used in this paper [29]. The pre-processing process loaded deactivated word lists and removed meaningless words, punctuation, special symbols, etc. Finally, the training set, test set, and validation set were divided according to the ratio of 8:1:1.

**Figure 8 ijerph-19-13520-f008:**
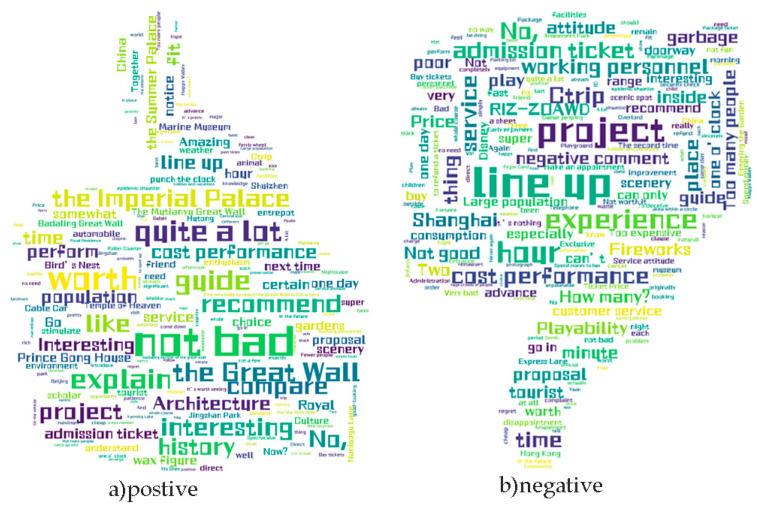
Word cloud display of positive and negative evaluations. Word clouds were created using the positive and negative reviews in the dataset to obtain the two highest rated word frequency images.

**Figure 9 ijerph-19-13520-f009:**
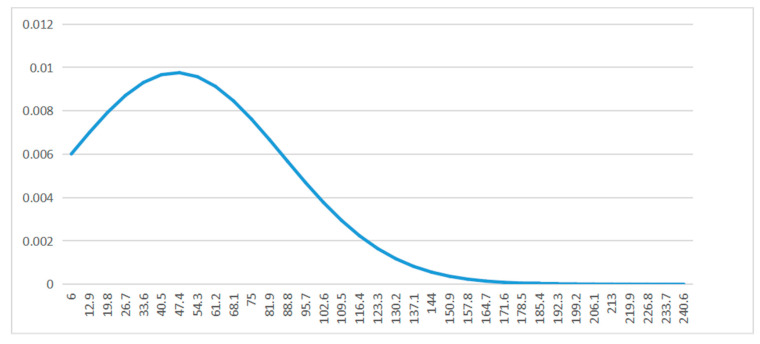
Distribution diagram of CAEIE. The mean length was 46.2. The maximum value was 351. The minimum value was 6. The standard deviation was 40.86. The group distance was 6.9. The number of groups was 50.

**Figure 10 ijerph-19-13520-f010:**
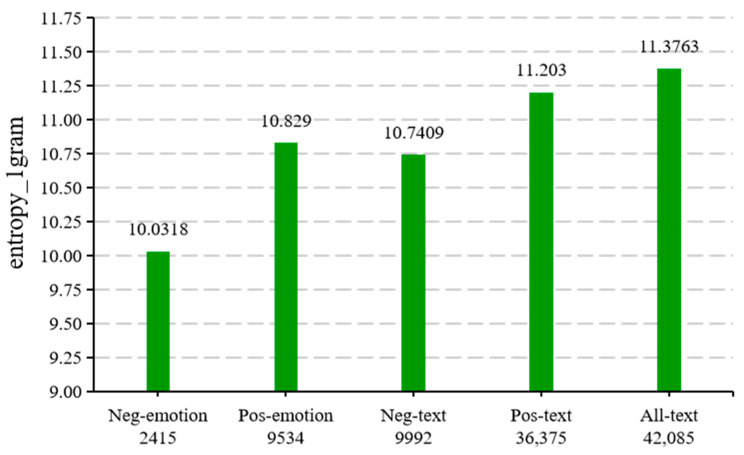
Information entropy of the dataset. The information entropy was calculated using Equation (19), which was the 1-g used in this paper. The numbers below the horizontal coordinates indicate the number of different words in each category. The number of different words in all-text was 42,085, and the information entropy was 11.3763 bits.

**Figure 11 ijerph-19-13520-f011:**
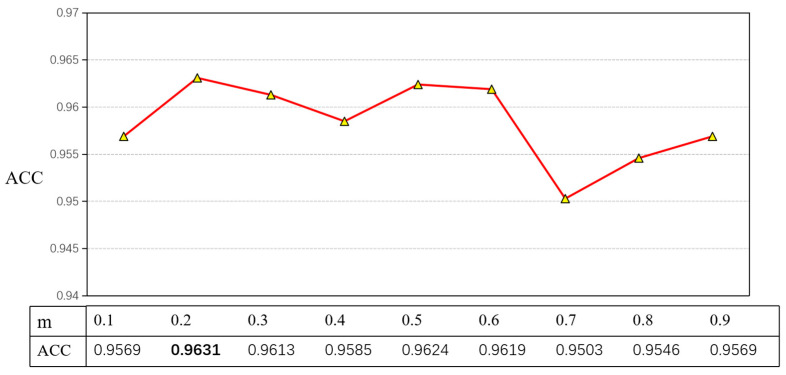
Testing the relationship between m and accuracy when GCN-layer = 1 and dropout = 0.1. The ACC reached the highest value of 0.9631. This was 0.07 higher than other values. The bolded font in the figure is the optimal effect.

**Figure 12 ijerph-19-13520-f012:**
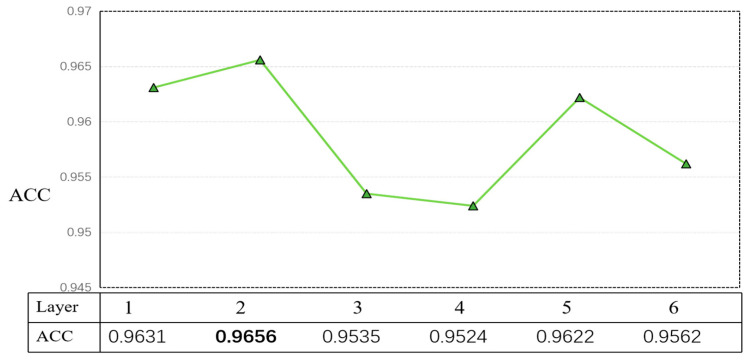
Relationship between GCN layer and accuracy, tested when m = 0.2 and dropout = 0.1 for GCN-layer. The results showed that the best results were obtained when GCN-layer = 2, leading by 0.25. The bolded font in the figure is the optimal effect.

**Figure 13 ijerph-19-13520-f013:**
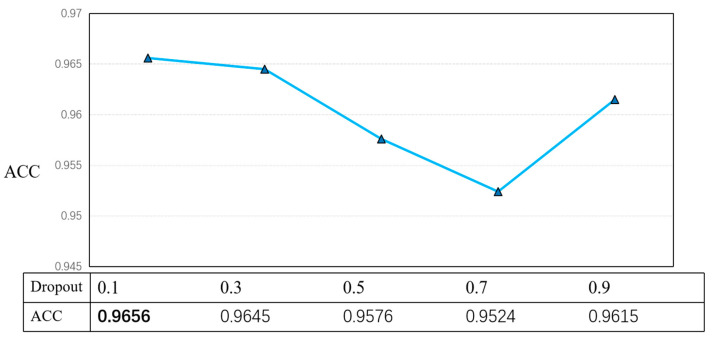
Test relationship between dropout and accuracy when GCN-layer = 2 and m = 0.2. The best result was tested for dropout = 0.1, leading by 0.11. The bolded font in the figure is the optimal effect.

**Table 1 ijerph-19-13520-t001:** Emotional emojis for labeling.

Classes	Emoji
Neg	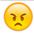 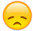 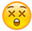 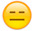 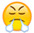 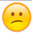 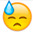 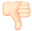
Pos	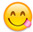 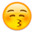 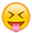 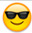 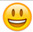 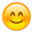 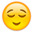 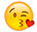 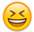 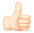 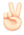 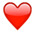

**Table 2 ijerph-19-13520-t002:** Chinese attractions evaluation incorporating emojis.

Classes	Review	Review-Translate
Emotion-Neg	这地方一年不如一年 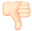 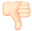 。	This place gets worse every year 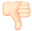 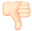 .
没去成，也不给退，闹过 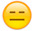 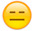 。	Didn’t go, also don’t give back, trouble 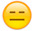 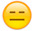 .
Emotion-Pos	整体感觉还行，玩不错，带着老人也可以去的地方 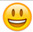 。	The overall feeling of this place is OK, very fun, with the old people can also go to the place 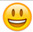 .
性价比还可以 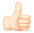 ，小孩子喜欢 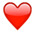 。	Cost performance is ok 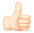 , children like it 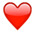 .
Text-Neg	景色一般，可玩性低，性价比低，有待改进。	General scenery, low playability, and low-cost performance are to be improved.
里面饭吃不起，外面饭带不进去。	It’s too expensive and you’re not allowed to bring your food.
Text-Pos	北京著名景点，到北京必去呦。	Beijing’s famous scenic spots, Beijing must go.
见到好多名人，栩栩如生，制作精致。	See a lot of famous wax figures, lifelike, exquisite production.

**Table 3 ijerph-19-13520-t003:** Chinese scenic spot evaluation text dataset with emotional symbols.

Classes		Train	Test	Dev	Total
Text	Neg	4489	561	561	5611
	Pos	26,567	3300	3300	36,197
Emoji	Neg	210	26	26	262
	Pos	2081	260	260	2601
Total		35,735	4468	4468	44,671

**Table 4 ijerph-19-13520-t004:** Confusion matrix.

Predict True	0	1
0	TP	FN
1	FP	TN

**Table 5 ijerph-19-13520-t005:** Performance of different models on CAEIE.

Model Name	Precision	Recall	F1	ACC
BERT	0.9724	0.984	0.9782	0.9599
Roberta	0.9715	0.9782	0.9748	0.954
ALBERT	0.8392	0.8602	0.8496	0.9588
ERNIE	0.9682	0.9815	0.9748	0.9537
ERNIE-Gram	0.9701	0.989	0.9795	0.9622
Skep	0.9732	0.978	0.9756	0.9567
TextGCN	0.9646	0.9892	0.9767	0.9572
Roberta+GCN	0.9738	0.9717	0.9727	0.9531
E2G	0.9797	0.99	0.9848	0.9736

**Table 6 ijerph-19-13520-t006:** Comparison of GAT and GCN on CAEIE.

Model Name	Precision	Recall	F1	ACC
Ernie+GCN	0.762	0.8805	0.8170	0.9494
Ernie+GAT	0.8846	0.8024	0.8415	0.9592
ERNIE-Gram+GAT	0.9221	0.7263	0.8126	0.9556
ERNIE-Gram+GCN	0.9797	0.99	0.9848	0.9736

**Table 7 ijerph-19-13520-t007:** Compare Different activation functions on CAEIE.

Activation Function	Precision	Recall	F1	ACC
ReLU	0.8321	0.8464	0.8392	0.9558
ReLU6	0.8531	0.8506	0.8518	0.961
ELU	0.8146	0.8636	0.8384	0.9549
Selu	0.7535	0.8837	0.8134	0.9453
CELU	0.8753	0.7912	0.8311	0.9583
Leaky ReLU	0.7959	0.8696	0.8311	0.9528
RReLU	0.8084	0.858	0.8325	0.9542
Gelu	0.6981	0.9102	0.7902	0.936

## Data Availability

The datasets CAEIE that were used in this study are available online on the following links: https://github.com/yyy4444/Chinese-attractions-evaluation-incorporating-emojis-.git (accessed on 24 September 2022).

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
