# Peer review of "Sentiment Classification of Chinese Tourism Reviews Based on ERNIE-Gram+GCN"

_ijerph, 2022, doi:10.3390/ijerph192013520_

Round 1

Reviewer 1 Report

The paper presents Chinese attractions evaluation incorporating emojis and Ernie-Gram and textGCN (E2G) model to classify the dataset with high accuracy.

Here are some comments for imporvements

1.       Which advanced models this paper is compared

2.       Why only CAEIE dataset is considered. Justification is required.

3.       What is ERNIE. There is nothing written about ERNIE, therefore, authors need to describe it

4.       There is no related work to compare with. Authors must write related work section

5.       Compare related work with other models and also list the clear differences between Chinese attractions evaluation incorporating emojis and published works.

6.       If there is no publicly available datasets, then how 40000 datasets are analyzed for results.

7.       I cant find the emoji in table 1.

8.       Figures 7-10 need to be explained well.

9.       Add future work in the conclusions section

Author Response

We much appreciate for your professional comments on our article. We have made corrections point by point as you suggested.

  1.      Which advanced models this paper is compared

answer:

In table1 of the experimental section of 5.5, the advanced models compared in this paper include BertGCN presented at ACL (a top conference in natural language) in 2021 and Ernie-gram presented at NAACL in 2021. And a set of comparison models SKEP is added in the paper. Skep is based on Roberta's model focusing on sentiment classification, which will be presented at ACL in 2020. In the text, a brief description of Skep is added at the top of Table 5 and the results of Skep on the CAEIE dataset (accuracy, precision, recall and F1) are shown in Table 5.

  1.      Why only CAEIE dataset is considered. Justification is required.

 answer:

There is no publicly available dataset of Chinese attraction reviews, much less one that takes emojis into account. This phenomenon is somewhat different from the current real reviews on the web. The rationale for using the CAEIE dataset is described at the end of the second paragraph of Chapter 2. The significance of the CAEIE dataset is added at the end of the last paragraph of the second chapter in the text.

  1.      What is ERNIE. There is nothing written about ERNIE, therefore, authors need to describe it

answer:

 ERNIE is based on Bert's model, which optimizes Bert's mask mechanism and is more suitable for Chinese. ERNIE is introduced in detail in 3.1.

  1.      There is no related work to compare with. Authors must write related work section

 answer:

The related work is integrated into the 1.introduction section. To address this point, it is proposed that the first part be split into 1.introduction and 2.related work. The serial numbers of the subsequent sections are added one after the other.

  1.      Compare related work with other models and also list the clear differences between Chinese attractions evaluation incorporating emojis and published works.

 answer:

The differences between the CAEIE dataset and other datasets are described at the end of the second section of the paper.

  1.      If there is no publicly available datasets, then how 40000 datasets are analyzed for results.

 answer:

In this paper, the information entropy of the dataset and the normal distribution of the length are added in 5.1 to prove the availability and diversity of the dataset. In 5.4, E2G is compared with several sets of advanced models. And the 10-fold Cross Validation method is used for validation, and the error of the final result is less than 0.001, which fully proves the performance of the E2G model.

  1.      I cant find the emoji in table 1.

 answer:

Due to a problem with the MDPI submission system, the emojis in Tables 1 and 2 could not be displayed in the submission system. The editor has been contacted to explain the situation. And the emojis have been reinserted into Table 1 and Table 2.

  1.      Figures 7-10 need to be explained well.

 answer:

Due to the insertion of two new figure between Figure 8 and Figure 9, an explanation of Figures 7 - 11 (now Figures 7, 8, 11, 12 and 13) was provided. A detailed explanation has been added below each image.

  1.      Add future work in the conclusions section

answer:

A description of the future work was added at the end of Chapter 6, covering two main points (1) reducing the time required for E2G training (2) identifying sarcastic comments in comments.

Reviewer 2 Report

The manuscript proposed a method for classifying sentiment with and without emoji based on ErniGram and TextGCN. The results show good result, however, there are several problems in the paper.

Please find attached annotated PDF file for detail.

Example of problems:

- The contribution needs to be explained shorter and clearer.

- Most figures are provided with low quality.

- Missing some contents, e.g. "emoji" in Table 1.

Author Response

We much appreciate for your professional comments on our article. We have made corrections point by point as you suggested.

1.Define ernieGram and TextGCN explicitly before or after mentioning it. It also applies for other terms and abbreviations.

answer:

The first occurrences of Ernie-Gram and TextGCN were given full names, and the full text was checked to give full names to all first occurrences of abbreviations such as ALbert, roberta, RELU, RNN, etc.

  1. experiments cannot compare, human can compare based on experiments.

answer:

The last sentence of the abstract has been changed to ’This paper compares the effects of eight activation functions on the second layer of the TextGCN, and the activation function RELU6 with the best results based on experiments.’

3.Three more paragraphs are started with the similar pattern: more/several scholars. Please rephrase it.

answer:

The relevant expressions in 2.related work have all been reworded.

4.This writing style is no appropriate. Combine the sentence "The main contribution .." into the las paragraph. Mention the contributions exactly as a list (short and clear).

answer:

In the last paragraph of Chapter 2, this paragraph has been adjusted due to a formatting error. And the main contribution is put in the last paragraph and the contribution is listed.

5.Remove this last from contribution. Explain as closing paragraph for the introduction.

answer:

Move the last point of contribution to the last paragraph of the introduction of Part I.

6.source of figure?

answer:

The image sources are labeled for Figures 1, 2, 3 and 4 in the text.

7.Need higher resolution. See that the image is blur when zoomed.

answer:

Fig. 4, Fig. 5 and Fig. 6 have been replaced, all with tiff files.

8.is this multiply or X?

answer:

In the formula section of 4.1(3), X in Eq. (18) denotes the node, and the specific formula is given in Eq. (14) above, and the explanation is added here in the text.

9.Emoji column is empty

answer:

Due to a problem with the MDPI submission system, the emojis in Tables 1 and 2 could not be displayed in the submission system. The editor has been contacted to explain the situation. And the emojis have been reinserted into Table 1 and Table 2.

Reviewer 3 Report

First of all congratulation on submitting the manuscript to the MDPI journal. To improve the quality of the manuscript the following suggestions could be considered:
1) I suggest avoiding such keywords as artificial intelligence, which is too general. Minor mistake line 20, before NLP, need to put space tab.
2) The Introduction section should be split into two sections: Introduction and Relates works.
3) After related works the novelty of this manuscript should be more highlighted, what the literature review has shown and what the authors will solve, what has been done differently.
4) I don't think the authors need to describe the all methods so deeply, because there is nothing new has been implemented in the steps, not modified, etc., so it would be enough a summary of the used methods, a few sentences, and differences between them. Now a lot of text is given, but nothing new.
5) The newly collected dataset should be presented more deeply, and the descriptive static should be presented: the distribution of the dataset according to the length of text, dataset class distribution, etc.
6) Usually the definition is "pre-processing", instead of pretreatment. Does only the stop words have been ignored, there other pre-processing filters have not been used? Like points, commas, or numbers, has been left? Which also will affect the results.
7) How the dataset has been labeled?
8) What is DEV in Figure 7? Usually, there are training, testing, and validation subsets.
9) How the dataset has been split into these subtests? If randomly, maybe the authors just were lucky, but if the experiments will be repeated, the results can be different. For such experiments, cross-validations would be more suitable.
10) Table 1 is empty.
11) I would suggest reconsidering of section 4.2., the information could be included in another section because one sentence section doesn't seem enough.

Good luck with the final submission.

Author Response

We much appreciate for your professional comments on our article. We have made corrections point by point as you suggested.

  • I suggest avoiding such keywords as artificial intelligence, which is too general. Minor mistake line 20, before NLP, need to put space tab.

answer:

Removed 'artificial intelligence' from the keywords and added a space before NLP.

2) The Introduction section should be split into two sections: Introduction and Relates works.

answer:

The original 1.introduction part has been split into 1.introduction and 2.related work

3) After related works the novelty of this manuscript should be more highlighted, what the literature review has shown and what the authors will solve, what has been done differently.

answer:

At the end of Section 2, the CAEIE dataset is described. And at the end, two innovative points of this paper are elaborated.

4) I don't think the authors need to describe the all methods so deeply, because there is nothing new has been implemented in the steps, not modified, etc., so it would be enough a summary of the used methods, a few sentences, and differences between them. Now a lot of text is given, but nothing new.

answer:

In this paper, the introduction of the method in the third part has been appropriately deleted.

5) The newly collected dataset should be presented more deeply, and the descriptive static should be presented: the distribution of the dataset according to the length of text, dataset class distribution, etc.

answer:

As shown in Figure 9, a distribution plot about the length of the dataset is added to better describe the distribution of the CAEIE dataset length.

6) Usually the definition is "pre-processing", instead of pretreatment. Does only the stop words have been ignored, there other pre-processing filters have not been used? Like points, commas, or numbers, has been left? Which also will affect the results.

answer:

Pre-processed before sorting. Load the deactivation word list, and the characters in the word list are cleared out. This word list includes punctuation, inflections, and special symbols, etc. A description was added to the bottom of Figure 7 in the text. The word 'pretreatment' was replaced with 'pre-processing' in the text and in the text.

7) How the dataset has been labeled?

answer:

This paper collects reviews on the Ctrip website, and the reviews themselves are scored by tourists. In this paper, 1-2 are classified as bad ratings, 3 are medium ratings, which are not used in this dataset, and 4-5 are positive ratings. This expression is illustrated below Figure 7 in the text.

8) What is DEV in Figure 7? Usually, there are training, testing, and validation subsets.

answer:

DEV is the validation set, change DEV to validation in Figure 7.

9) How the dataset has been split into these subtests? If randomly, maybe the authors just were lucky, but if the experiments will be repeated, the results can be different. For such experiments, cross-validations would be more suitable.

answer:

In this paper, a 10-fold cross-validation was conducted. The results of the cross-validation showed that the E2G effect reached the highest, with an error of no more than 0.001. table 5 This note was added in the first paragraph below.

10) Table 1 is empty.

answer:

Due to a problem with the MDPI submission system, the emojis in Tables 1 and 2 could not be displayed in the submission system. The editor has been contacted to explain the situation. And the emojis have been reinserted into Table 1 and Table 2.

11) I would suggest reconsidering of section 4.2., the information could be included in another section because one sentence section doesn't seem enough.

answer:

Due to the addition of previous chapters, the original 4.2 became 5.2. In this paper, 5.2 is put into 5.3 experimental details, and the 5.2 serial number becomes 5.2.1.In addition, 5.3 becomes 5.2.2 (parameter setting).

Reviewer 4 Report

The manuscript reports on a production of a dataset of text-based tourism reviews in the Chinese language and an approach to sentiment classification of these reviews based on Ernie-gram and text graph convolutional networks. In terms of topic, it is suited to the Journal. In terms of methodological approach and presentation, the manuscript suffers from certain drawbacks: The authors are advised to consider the possibility to address the following remarks.

Remarks:

 - Introduction:

1. The paragraph of Introduction starting at line 117 and ending at line 125 is especially important in announcing the methodological contribution of the represented study. However, this paragraph is too concise and needs additional clarifications. The authors should explain the following notions in more details:

1.1 The authors represent these notions of “overall information of the dataset” and “the contextual features” (cf. p. 3,  l. 118-119) as being contrastive, but do not explain the dichotomy between them. Both notions should be explained in more details.

1.2. Similarly, “the relationship between coarse-grained and fine-grained” (cf. p. 3,  l. 120-121) should be clarified or exemplified (e.g., explain the notions of coarse-grained and fine-grained units, etc.).

 - Methodology:

2. Section 2 should make a more clear difference between the authors’ methodological contribution and other researchers’ work. Make a clear difference between other people work and the methodological contribution of the authors.

 - Dataset:

3. Several important questions related to the production of the dataset should be addressed: Why emoticons were added to the dataset by authors? How the “reasonable positions” for added emoticons were determined? What is the size of the dataset part to which emotions were added by authors? And most importantly: how did the authors ensure that their (external) insertion of emoticons do not affect the representativeness and balance of the dataset.

4. The authors interchangeably and inconsistently use the terms “emoticon” and “emoji” through the manuscript (to illustrate this remark: the caption of Table 1 contains “emoticons”, while the header of second column of this table contains “emoji”, etc.). The authors should choose appropriate term and use it consistently.

5. Table 1 does not contain any emoticons.

6. Table 2 is described as including “reviews containing emojis” (cf. p.11, l. 377). However, no emojis/emoticons are contained in the given reviews.

7. The authors state that “Figure 8, it is easy to conclude that the keywords for good reviews in a)
are: very good, go again, recommendable, good, etc. In b), the keywords for bad reviews are queue, service, guide, crowded, poor experience, etc.”. This conclusion is not really supported, because words clouds reflect word frequency. If the authors want to provide a subset of (most) discriminating words, the selection should be based on information gain of individual words (based on information entropy).

 - Results:

8. For the purpose of result evaluation, it would be useful if the authors provided additional standard parameters - besides accuracy (cf. Eq. (1), p. 14) -  such as: precision, recall and balanced F-1 measure.

Additional remarks:

9. Incorrect section numbering: There is no Section 3.

10. Reference “#9”? (cf. p. 14, l. 429)

11. There are two same consecutive sentences (cf. p. 14, l. 432).

Author Response

We much appreciate for your professional comments on our article. We have made corrections point by point as you suggested.

  1. The paragraph of Introduction starting at line 117 and ending at line 125 is especially important in announcing the methodological contribution of the represented study. However, this paragraph is too concise and needs additional clarifications. The authors should explain the following notions in more details:

1.1 The authors represent these notions of “overall information of the dataset” and “the contextual features” (cf. p. 3,  l. 118-119) as being contrastive, but do not explain the dichotomy between them. Both notions should be explained in more details.

answer:

An explanation is added later in the section that explains the“overall information of the dataset” and “the contextual features”.

1.2. Similarly, “the relationship between coarse-grained and fine-grained” (cf. p. 3,  l. 120-121) should be clarified or exemplified (e.g., explain the notions of coarse-grained and fine-grained units, etc.).

answer:

The relationship between the two is explained after “the relationship between coarse-grained and fine-grained” .

  1. Section 2 should make a more clear difference between the authors’ methodological contribution and other researchers’ work. Make a clear difference between other people work and the methodological contribution of the authors.

answer:

At the end of Section 2, above 'Our contribution', the advantages of the E2G model over other models are described and the advantages of the CAEIE dataset are elaborated.

  1. Several important questions related to the production of the dataset should be addressed: Why emoticons were added to the dataset by authors? How the “reasonable positions” for added emoticons were determined? What is the size of the dataset part to which emotions were added by authors? And most importantly: how did the authors ensure that their (external) insertion of emoticons do not affect the representativeness and balance of the dataset.

answer:

The three recommendations above are supplemented in the paragraphs above Table 1.

  1. Since some emojis contained in the collected dataset are unique to Android or Apple systems. This part of the emojis cannot be displayed in windows and Linux, and cannot be encoded and decoded. We add the emojis that can be displayed to the original position of the non-displayable symbols (with the same sentiment).
  2. Some of them are direct replacements for symbols that cannot be displayed. Another part is that we observed the location of emoticons in real-life evaluations and found that most of them are at the end of a sentence, and that people will type more than one symbol when they are happy or angry. We followed this phenomenon to expand the dataset appropriately.

3.The part of the added sentiment dataset is about 1200 items. In the collected dataset, the medium reviews are discarded. However, when including the medium reviews, the ratio of plain text and reviews with emojis in the overall dataset can be about 9:1. after discarding the medium reviews, a large number of reviews with emojis are discarded. We added them manually to compensate for this part and maintain the overall ratio still at 9:1.

  1. The authors interchangeably and inconsistently use the terms “emoticon” and “emoji” through the manuscript (to illustrate this remark: the caption of Table 1 contains “emoticons”, while the header of second column of this table contains “emoji”, etc.). The authors should choose appropriate term and use it consistently.

answer:

Replace all "emoticon" with "emoji" in the original text.

  1. Table 1 does not contain any emoticons.

answer:

Due to a problem with the MDPI submission system, the emojis in Tables 1 and 2 could not be displayed in the submission system. The editor has been contacted to explain the situation. And the emojis have been reinserted into Table 1 and Table 2.

  1. Table 2 is described as including “reviews containing emojis” (cf. p.11, l. 377). However, no emojis/emoticons are contained in the given reviews.

answer:

Due to a problem with the MDPI submission system, the emojis in Tables 1 and 2 could not be displayed in the submission system. The editor has been contacted to explain the situation. And the emojis have been reinserted into Table 1 and Table 2.

  1. The authors state that “Figure 8, it is easy to conclude that the keywords for good reviews in a)

are: very good, go again, recommendable, good, etc. In b), the keywords for bad reviews are queue, service, guide, crowded, poor experience, etc.”. This conclusion is not really supported, because words clouds reflect word frequency. If the authors want to provide a subset of (most) discriminating words, the selection should be based on information gain of individual words (based on information entropy).

answer:

As shown in Figure 10, the information entropy of the four categories in the dataset and the overall dataset containing, favorable and unfavorable reviews with expressions, and favorable and unfavorable reviews without expressions, and the dataset as a whole were added for calculation. The bottom of Figure 10 indicates the category and the number of different words in that category. The 1-gram calculated in this paper.

  1. For the purpose of result evaluation, it would be useful if the authors provided additional standard parameters - besides accuracy (cf. Eq. (1), p. 14) -  such as: precision, recall and balanced F-1 measure.

Three evaluation criteria were added to this paper. The corresponding equations were added in 5.3. And the experimental results are added in Tables 5,6 and 7. And additional explanations are provided at the bottom of Table 5 ,Table 6 and Table 7.

  1. Incorrect section numbering: There is no Section 3.

answer:

After inspection and modification, there are now six chapters in total. 1. Introduction 2. Related work 3. Methods 4. E2G 5. Experimental part 6. Conclusion

  1. Reference “#9”? (cf. p. 14, l. 429)

answer:

A revision has been made for this reference and all reference citations throughout the text have been checked.

  1. There are two same consecutive sentences (cf. p. 14, l. 432).

answer:

The repeated sentences are deleted.

Round 2

Reviewer 3 Report

Thank you for taking the suggestions into account. The manuscript has been improved and could be published in the journal. Good luck with the final submission.

Reviewer 4 Report

The authors have adequately addressed most of the remarks from my previous review report and I believe that the revised manuscript has been sufficiently improved to warrant publication in IJERPH.

Minor remark: The paper would benefit from English proofreading.